# The Influence of Romanian Mobile Commerce Companies on Developing Green Innovation

**Violeta Mihaela Dincă [1], Anca Bogdan [1], Cristinel Vasiliu [2,\*] and Francisca Zamfir [3]**

1 UNESCO Department for Business Administration, Faculty of Business Administration in Foreign Languages, Bucharest University of Economic Studies, 010375 Bucharest, Romania; violeta.dinca@fabiz.ase.ro (V.M.D.); anca.bogdan@fabiz.ase.ro (A.B.)

2 Department of Business, Consumer Sciences and Quality Management, Faculty of Business and Tourism, Bucharest University of Economic Studies, 010374 Bucharest, Romania

3 Doctoral School in Business Administration, Bucharest University of Economic Studies, 010371 Bucharest, Romania; francisca.zamfir@fabiz.ase.ro

\* Correspondence: cristinel.vasiliu@com.ase.ro; Tel.: +40-766-546-221

**Abstract:** Lately, concern for green innovation has expanded in the business environment and many companies see it as a helpful element to gain competitive advantage. Due to the strains of maintaining sustainable businesses, mobile commerce companies are propelled to build up their own green innovation programs and harmonize them with the firm's management programs. The central scope of this research is to examine the drivers for green innovation within a range of Romania-based mobile commerce companies that operate in different industries. With the aim of identifying the factors that determine the development of green innovation, a conceptual model has been conceived. The dependent variable within the model is the action of the company's management to develop green innovation. Four independent types of variable structures that have an impact on the development of green innovation in mobile commerce firms were distinguished. The four categories of constructs are business environmental factors (1), green training (2), green supplier development (3), and technological factors (4). An online survey tested the model based on the responses of senior level management representatives from 182 Romanian mobile commerce companies from Bucharest. The validity of the model was fulfilled though factor analysis and reliability tests for the data; a logistic regression analysis was also used to test the research hypotheses. The research revealed that green training embodies the fundamental element enhancing green innovation among Romanian mobile commerce companies. This article benefits both academia and business. Firms could be inspired by the results of this paper to broaden the level of green innovation throughout the Romanian business environment.

**Keywords:** green innovation; mobile commerce; Romanian firms; theoretical model

## 1. Introduction

Recently, interest in green innovation has been growing in the business world since it is considered a strategic priority to gain competitive advantage. Instead of being considered expensive, green measures have turned into a basis for competitiveness [1]. Green innovation is described as the development of environmentally friendly products and processes [2] by means of implementing organizational practices like greener raw materials, using fewer materials within the process of products' design, using ecodesign principles with the aim of reducing emissions, and reducing the consumption of water, electricity, and other raw materials [3]. Certain prior studies advanced the idea that organizations which show green innovativeness hold higher achievements and gain more noteworthy accomplishments than competitor companies, because they exploit their green resources and skills to respond to clients' necessities quickly and suitably [4] and contribute intangible values into the organization. Different prior surveys advanced that the idea that human

resource management favorably and substantially impacts technological and product innovation [5,6], confirmed recently also by the study of Singh, Del Giudice, Chierici, and Graziano [7].

When focusing on the consumer experience in relation to the topics mentioned above, the present-day consumer is more and more interested in a positive environmental impact. The main problem statement of this article combines two innovative ecommerce trends for the decade of the 2020s. The first trend consists of the fact that mobile ecommerce has grown in the last five years to more than 70% of the EU's ecommerce sales [8] and therefore enhancing the ecommerce experience of mobile users could be a great opportunity for businesses, which optimize their websites and online stores for mobile use, making traditional "eCommerce" accessible to a larger audience. For ecommerce, mobiles shaped the way brands approach consumers but also the behavior of consumers towards businesses and products. Sometimes defined as "a build-up of e-commerce" [9] or as "a wider notion applied for mobile banking, mobile ticketing, mobile coupons, procuring goods and services by the means of mobile phones" [10], m-commerce has established itself as a milestone novelty of mobile technology capitalizing on a landscape of dealing with purchases, payments, and exchanges wirelessly [11].

The second trend is green consumerism which is transforming and tailoring the ecommerce area, and many businesses have adapted to this trend and continue to do so. Since the number of consumers who take into consideration environmental issues when judging their shopping selections and who embrace more sustainable practices in terms of what they choose to consume is growing, businesses need to configure innovative methods to make their operations environmentally friendly. In particular, the category of millennial consumers is strongly modifying the green economy through their buying an increasingly complex assortment of green products ranging from vegan cosmetics to organic foods, because this category of consumers is gaining a feeling of being responsible for the planet. Therefore, it is essential for mobile commerce businesses to foster an ecofriendly strategy that caters to the needs of these clients [12].

These two trends have been evolving in Romania as well as in other countries of the European Union. On one hand smartphones have become the main support for the development of retailers' multichannel strategies in Romania for 2021. Prior to the outbreak of COVID-19, Romanian consumers would commonly browse for information on desired products on their mobile phones, while finalizing purchases in physical stores but during the crisis smartphones moved from browsing to purchase completion [13]. On the other hand, since the Romanian economy is at the same time a main consumer of natural resources and a major producer of environmental impacts, exploring more sustainable alternatives is the only way to decrease the environmental impact of a company [14] and mobile commerce is a very suitable and fitting business area in which to take steps towards green innovation. Therefore, the main objective of this research was to investigate the factors having significant influences on the development of green innovation in the mobile commerce firm context for Romania. The paper describes the most important studies on the topic of green innovation and their results, gathered in the literature review section, followed by the hypotheses formulation with the described studies as backbone. The practical approach of the paper is given in the next section (methodology) and includes the conception of a model reuniting the factors with the potential to influence the green innovation of firms. This model is assessed through a quantitative approach with an online survey methodology to gather the data necessary for testing the hypotheses on 182 Romanian mobile commerce companies. The results cover the outcomes obtained through the descriptive and factor analysis and logistic regression applied on the model.

## 2. Literature Review and Hypotheses Formulation

Many researchers described ecoinnovation or green innovation as innovative programs integrated within the environmental management of a company [15]. Green innovation is defined as the advancement of services, modern methods, and a life cycle that can

lead to the decrease in environmental menaces, pollution, and other unfavorable effects of resource use in connection with other options [16,17].

Ecommerce companies have revolutionized the way the whole world does business because they have built up comfort and ease for consumers, making it possible for them to buy what they desire, whenever and wherever they want. At the same time, they have created broader cutting-edge chances and options for sellers, distributors, and tradespeople overall but also brought extra defiance and demanding tasks, especially for the sphere of sustainable development. Even though it sounds like a difficult challenge, according to a US study performed in 2019 regarding the most important features when choosing an online commerce to shop, 71% of consumers claimed that a firm's duty and dedication to environmental issues was a primary interest, 68% affirmed that the firm's social input was of high importance, and 65% of consumers said that it was of great significance for them if the firm was involved in charity/social responsibility actions in the community [17].

Sustainability has therefore become a major concern for online shopping companies and since online commerce is gaining more significant weight for the Romanian economy as well, it is important to examine how these two concepts will evolve together. Ecommerce companies can take steps to make Romanian consumption more sustainable by reducing waste, getting customers involved, and training their employees in sustainable practices. [18].

Taking elements from the above mentioned theories and studies, a conceptual model with multiple hypotheses was developed. In the research model which was put forward, the essential emphasis was on identifying the drivers for green innovation within Romanian mobile commerce firms.

The first direction taken was to examine ways that government policies could encourage firms to increase their green innovation. One example of such an action would be that if companies do not have sufficient incentives to carry out research and development, one possibility would be for the government to fund such work directly. Another example would be to encourage cooperative research ventures between universities and companies. Regarding government input for developing green innovation, there is a certain shortage of work in literature. One solid study on the matter provided by Bai, Song, Jiao, and Yang (2019) confirmed that government R&D subsidies significantly increased the green innovation tendency and performance of Chinese commerce firms with the impact being stronger in state-owned enterprises and in small and medium enterprises [19]. Because we considered that the relationship between government support and green innovation has not been sufficiently researched and because mobile commerce has arrived in the ecommerce category, Hypothesis One is proposed: Government support for mobile commerce firms is positively related to the implementation of green innovation (Hypothesis One).

Based on the literature, the attitude of consumers towards environmental products has increased along with their care for green products. A recent study carried out by Aibar-Guzman and Somohano-Rodriguez (2021) on 5391 international companies corresponding to the period 2002–2017 showed that proactive environmental innovation strategies are positively valued by consumers, having a positive impact on companies' sales growth [20]. However, many studies focus on the fact that product innovation is appreciated by customers, and there are other important aspects for a business to improve, especially in ecommerce or mobile-commerce (which are not manufacturers so product innovation does not apply for them) such as improving shipping conditions, changing packaging, partnering with an environmentally friendly delivery company, or optimizing the company's returns strategy. It would be interesting to see if these practices are also appreciated by customers. Some studies, for example, mention that it is important to find green ways to improve the direct-to-consumer brand's largest and most important asset–its online storefront and draw attention to the fact that depending on the hosts of the online stores, the companies could be generating a sizeable carbon footprint, and switching to a green web-hosting firm is a straightforward way to instantly reduce the carbon footprint [21,22]. Therefore, Hypothesis Two is advanced: customer and competition

pressure on mobile commerce firms is favorably connected with the development of green innovation (Hypothesis Two).

The literature on the topic offers a wider view of the next factor that was taken into consideration for the model, which is related to green human resources management practices that can contribute to green innovation. Among them, green or environmental training (GT) is highlighted many times [23]. GT can be characterized as a method of on-the-job training and continued education designed to fulfill corporate environmental management goals and ambitions [24]. Following the studies of Paille et al. (2014) and Muduli et al. (2013) [25,26], GT is a form of training connected to pertinent environmental subjects; it facilitates all workers to fuse the company's performance with solving environmental problems. Past research argues that GT is positively related to the greening of companies around the world, for example, Sarkis et al. (2010) [27] asserted that GT contributed to integrating complex environmental practices among organizations from Spain. Jabbour (2015) also demonstrated that GT is positively correlated to the development of environmental management on a corporate level [28,29]. As a result, it is firmly assumed that green training in ecodesign, LCA, recycling/reusing, and waste elimination will point the way to green innovation advancement [30,31]. Because of all the above, Hypothesis Three is set forward: mobile commerce companies' green training implementation is positively related to their green innovation.

Islam, Karia, Fauzi, and Soliman (2017) [32] stated that, initially, the literature on green supply chain development was limited to a few aspects such as green purchasing, green packaging, and green manufacturing. With time, the green supply chain development discipline has been gradually growing, more and more researchers are expanding the work in different sub disciplines and the steps taken towards strengthening green suppliers actually can relate to a higher number of green innovations ([33]—Chay). Green supplier development consists of the actions taken by the firm with suppliers that foster and advance green performance such as (a) working together rather than halting the relationship with the suppliers in case of unsatisfying green performance, (b) visiting supplier plants and supporting them to upgrade environmental performance, (c) promptly and regularly being in contact regarding green performance issues, and (d) recognizing green supplier performance, e.g., through prizes or rewards, and close cooperation with suppliers on green topics [34,35]. Therefore, Hypothesis Four is submitted: green supplier development programs that are implemented by mobile commerce firms are positively related to the firms' green innovation.

Another set of notable determinants of innovation implementation is the influence of technological factors. A relevant study by Jumadi and Zailani (2011) affirmed that the explicitness and accumulation of technology might influence the development of green innovation [36]. According to the research conducted by Lin and Ho (2010), the technological factors with an impact on green innovation include relative advantage, compatibility, and complexity of green practices [37]. A challenging and complex technology encloses a large amount of tacit knowledge that involves work to acquire and disperse [38,39]. The issue of studying and diffusing tacit technological knowledge complicates further the process of advanced technology adoption. On the other hand, compatibility represents the level to which an innovation is seen as being relatable to the current values, practices, and needs of the companies [40,41]. Green innovations that are more suited to a firm's existing technologies will be shared with more ease within the company. Lastly, relative advantage can be defined as the notion that an innovation is more profitable than its alternative idea. The discerned advantage may be scaled at economic and social levels such as accessibility and satisfaction [42,43]. Therefore, Hypotheses Five through Seven are proposed: there is a negative link between the perceived green innovation's complexity and green innovation development for mobile commerce firms (H.5); there is a positive link between the perceived green innovation's compatibility and the actual green innovation implementation process within mobile commerce firms (H.6); there is a positive bond between the perceived green innovation's relative advantage and green innovation devel-

opment for mobile commerce firms (H.7). A clearer background of the research variables, their descriptions, and the hypotheses formulated is provided below by Table 1.

**Table 1.** Factors Influencing the Adoption of Green Innovation by Mobile commerce companies.

| Variable Name | Explanation of Variable | Formulation of Hypothesis (with Emphasis on the Effect of the Variable on the Decision of Implementing Green Innovation) |
|---|---|---|
| 1. Government contribution for the green innovation of firms | Government subsidies/interventions in supporting firms to adopt/implement green innovation and environmental protection | H1: More solid support from the government positively affects the chances for green innovation development within mobile commerce companies. |
| 2. Customers and competitors' influence | The importance that customers and competitors give to green innovation implementation | H2: Mobile commerce firms that are involved within a business environment strongly influenced by the customers' and competitors' feedback are more likely to develop green innovation. |
| 3. Green training programs | Organizing training programs for employees | H3: Mobile commerce companies that develop solid green training programs are more likely to enhance green innovation. |
| 4. Green supplier development programs | Integrating environmental thinking into supply chain management | H4: Mobile commerce firms that strengthen green supplier development programs are more credible to advance green innovation. |
| 5. Complexity of green innovation | The degree to which green innovation is perceived as being relatively difficult to understand and use | H5: Green innovation's perceived complexity negatively affects green innovation development for mobile commerce firms. |
| 6. Compatibility of green innovation | The degree to which green innovation is perceived as consistent with the existing values, past experience, and needs of companies | H6: Green innovation's perceived compatibility positively influences the green innovation implementation process within mobile commerce firms. |
| 7. Relative advantage obtained by the firm from implementing green innovation | The degree to which decision makers perceive green innovation as being more useful than other paradigms | H7: Green innovation's perceived relative advantage positively impacts green innovation development for mobile commerce firms. |

## 3. Methodology

In order to achieve the main objective of this research, which is to investigate the drivers for green innovation within Romanian mobile commerce companies, a conceptual model with four constructs was designed based on the seven hypotheses. The model was then assessed through a quantitative approach with an online survey methodology to gather the data necessary for testing the hypotheses. Surveys are suitable for collecting a substantial number of answers relatively economically [44,45], as well as for permitting the quantification of answers and statistical testing of the validity of the achieved outcomes, and the exactness of survey data relies upon the quality of the sampling methods used.

All data were gathered and investigated at firm level. A total of 182 Romanian based mobile commerce firms were contacted with the goal of collecting the data between February and April 2021. All 182 companies already provide a range of green products to their



consumers. The purpose of the article is to go further than offering green products to the public and investigate green innovation within the processes and activities of the company in a multifaceted way. The sample of survey respondents was chosen from senior management representatives of the mobile commerce companies such as: general manager, content development manager, catalog managers, supply chain manager, packaging division manager, customer support manager, marketing manager, and HR manager. The data collection process was founded on a questionnaire with the goal of recording the respondents' judgements on what green innovation elements (factors) could persuade them to implement it in their respective firms. The quantification tool used for the respondents' answers was the 5-point Likert scale analyzing the influence of all factors on implementing green innovation within the company (with 1 as the factor with the least influence and 5 as the largest influence). Further explanation of the research is given in the diagram below (Figure 1).

**Figure 1.** Research diagram of the study.

The quantitative analysis of the collected data was fulfilled by using a particular statistical software program (SPSS version 20). The validity of the model was explored by performing factor analysis and subsequently a logistic regression analysis was set up to test the research hypotheses. This method offered the opportunity to predict the determinants which were meaningful in enhancing the green innovations processes.

The construct validity method permits correlation analysis between items and the development of a new set of variables which are highly correlated between them, these new variables being called factors. Convergent validity is a constituent piece of construct validity that shows whether all the elements which affect a factor converge. A way to assess the convergent validity of the construct is the method of factor loading. Numerous studies showed that factor loadings need to be larger than 0.5 in order to deliver more accurate results [46,47], whereas Chen and Tsai (2007) acknowledged as well that 0.5 is a good value for acceptable loadings [48]. Additionally, while exploring pro-environmental consumer behavior, Ertz, Karakas and Sarigollu (2016) considered the factor loadings of 0.4 and above for their confirmatory factor analysis [49]. Carmines and Zeller (1979) suggested that the acceptable threshold for factor loadings is 0.7 or above to make the analysis more feasible [50]. The next step of construct validity that needs to be calculated is discriminant validity, which represents the span across which a construct is distinct from other constructs by empirical standards. Cross-loadings are a technique for estimating discriminant validity for reflective models. When investigating cross-loadings, each item's outer loading on a construct needs to be larger than all its cross-loadings with other constructs; otherwise, it means that the respective item does not really measure only one factor but more, and the model has weaknesses [51]. With the testing discriminant and convergent validity of the model, factor analysis was fulfilled to lessen the number of factors and to remove the items which had high cross loadings.

Next, logistic regression was applied to test the hypotheses and anticipate the evolution of green innovation within the companies. The regression's coefficients showed how

the constructs impacted the dependent variable. For this research, the dependent variable was binary (a mobile commerce firm either takes measures for green innovation or not) and the process of logistic regression was carried out to reveal the criteria which influenced green innovation progress within the respective company.

## 4. Descriptive Analysis, Hypothesis Testing, and Results

### 4.1. Descriptive Analysis

This section reveals the results of the descriptive analysis, which was applied to outline all the variables of the model precisely. The features of the respondents (decision makers) and the firms they represent can be seen in Table 2. In this subsection, the descriptive analysis of the variables is included (it also represented a basis for the factor analysis test encompassed in the next subsection).

**Table 2.** Sample's features.

| Features of the Sample | Number | Features of the Sample | Number |
|---|---|---|---|
| *Responder's job title* | | *Company's size by number of employees* | |
| General manager | 23 | 1–50 Employees | 25 |
| Content development manager | 31 | 51–100 Employees | 111 |
| Catalog manager | 27 | 101–250 Employees | 46 |
| Supply chain manager | 29 | Total | 182 |
| Packaging division manager | 16 | *Main Product/ Service category of the mobile commerce firm* | |
| Customer support manager | 30 | Electro/ IT&C | 39 |
| Marketing manager | 12 | Fashion | 18 |
| HR manager | 14 | Home & Garden | 26 |
| Total | 182 | Beauty | 17 |
| *Responder's years of experience within the firm* | | Pet Shop | 20 |
| <5 years | 48 | Books, Music & Movies | 14 |
| 6–10 Years | 103 | Auto & Moto | 25 |
| >10 Years | 31 | Articles for children | 23 |
| Total | 182 | Total | 182 |

In Table 3, the descriptive analysis of the items assessing the government's involvement in the development of green innovation within the firm is showed. This variable is scaled around the level of importance set by the respondent on how each item can constitute a catalyst for green innovation: the financial support provided by the government for adopting green practices (FinSupGov), the technical assistance provided by the government for adopting green practices (TechAssGov), and the environmental regulations set by the government for business operations (EnvRegGov). At the same time Figure 2 shows the answers of the respondents to each of the three items revealing that more than 50% agreed and strongly agreed that government support is necessary and beneficial for the development of green innovation within the mobile commerce company.

**Table 3.** Descriptive analysis for the variables of government contribution to green innovation.

| | N | Mean | Std. Deviation |
|---|---|---|---|
| FinSupGov | 182 | 4.39 | 0.814 |
| TechAssGov | 182 | 4.14 | 0.853 |
| EnvRegGov | 182 | 4.23 | 0.927 |

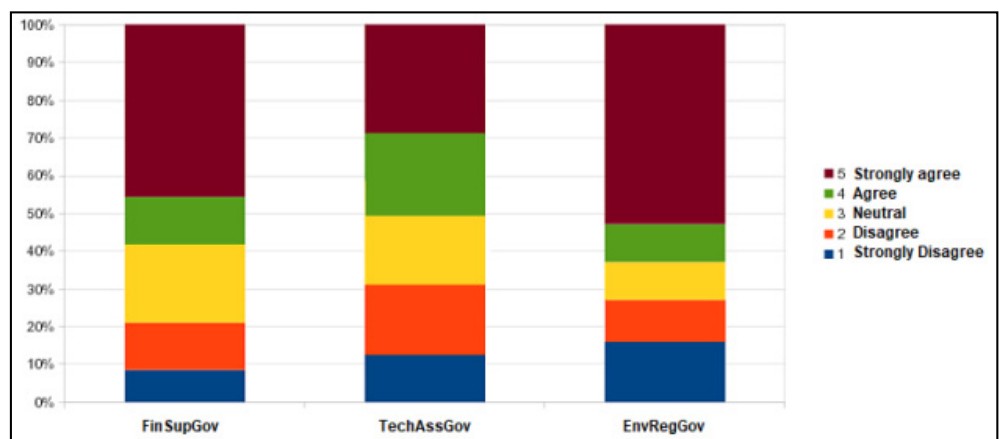

**Figure 2.** Points of view from the respondents regarding the contribution of the government to the development of green innovation.

The factors related to the influence of the customers and competitors on the company's green innovation are presented in Table 4 as follows: the pressure set by the customers, which requires that the companies improve their environmental performance (PressCustomEnvPerf), the emphasis that the customers place upon the company caring and being concerned for the environment (CustomCareEnvin), the difficulty of predicting competitors' behavior (PredCompet), the difficulty of predicting customers' preferences (PredCustom), the frequency of the changes of customers' preferences (FreqCustomPref). It needs to be mentioned that over 60% of the respondents agreed and strongly agreed that customers and competitors affect the development of green innovation within the mobile commerce firm (Figure 3).

**Table 4.** Descriptive analysis for the variables of customers and competitors' influence.

|  | N | Mean | Std. Deviation |
|---|---|---|---|
| PressCustEnvPerf | 182 | 4.37 | 0.841 |
| CustomCareEnvin | 182 | 4.41 | 0.826 |
| PredCompet | 182 | 4.22 | 0.804 |
| PredCustom | 182 | 4.04 | 0.801 |
| FreqCustomPref | 182 | 4.21 | 0.937 |

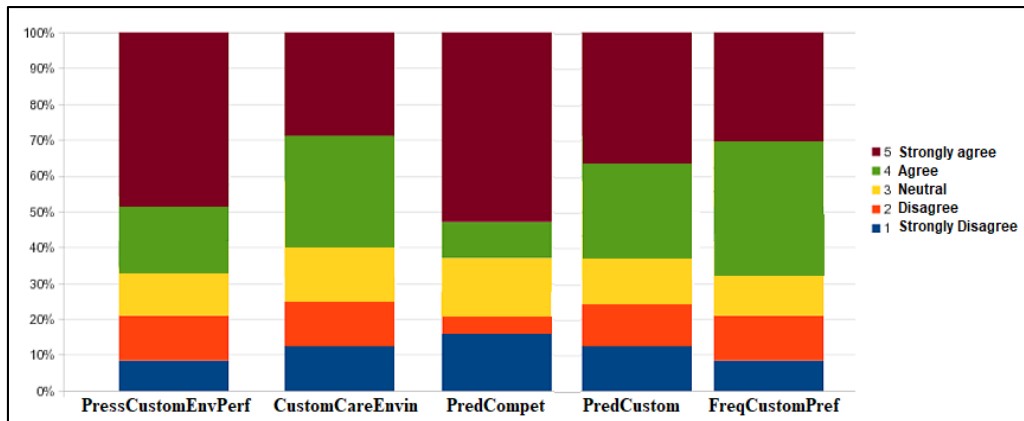

**Figure 3.** Points of view from the respondents regarding the influence of the customers and competitors on the development of green innovation.

The green training input on green innovation was assessed taking into consideration the respondent's evaluation of: the frequency of the training provided to the employees on methodologies and procedures for ecodesign, LCA, recycling/ reusing of materials, and disposal of waste (TrainProc), the sufficiency of the amount of trainings on environmental issues provided to the employees (SuffTrain), the existence of opportunities for environmental training participation for all employees (ExistTrain), the high level of the environmental training content delivered to the employees (HighLevContTrain), the efficiency of using the environmental training efficiently afterwards (EfficTrain). Table 5 reviews the mean and the standard deviation of these five items while Figure 4 shows that more than 70% of the representatives of the participant companies agreed or strongly agreed that green training is a key factor for boosting green innovation.

**Table 5.** Descriptive analysis for the variables of green training.

|  | N | Mean | Std. Dev. |
|---|---|---|---|
| TrainProc | 182 | 4.78 | 0.925 |
| SuffTrain | 182 | 4.95 | 0.939 |
| ExistTrain | 182 | 4.53 | 0.910 |
| HighLevContTrain | 182 | 4.72 | 0.903 |
| EfficTrain | 182 | 4.61 | 0.917 |

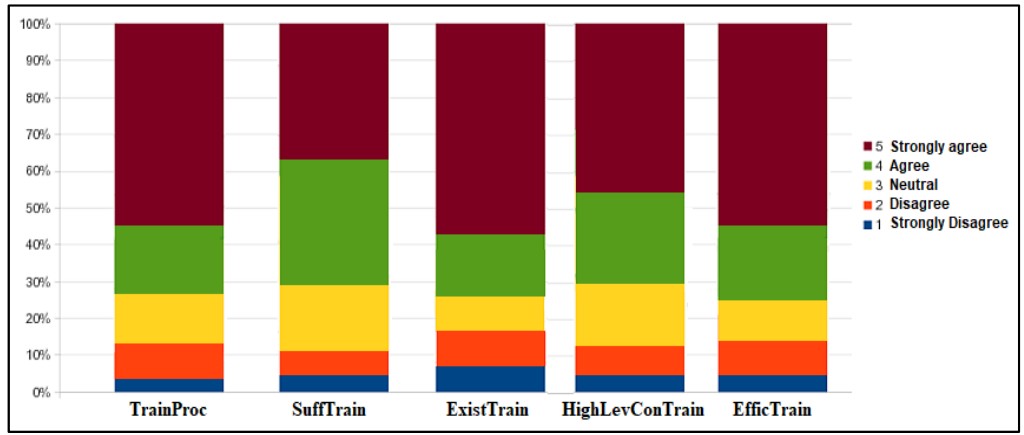

**Figure 4.** Points of view from the respondents regarding the influence of green training on the development of green innovation.

Green supplier development was measured by the respondent's point of view regarding: the regular and frequent communication with the suppliers on sustainability related issues (CommSupplSust), the engagement in close cooperation with a limited number of suppliers for the greening of the company's supply chain (CoopLimSuppl), the choice of developing instead of abandoning suppliers if they fail to meet environmental standards (ChoiceDevSuppl), involvement with the supplier in eliminating nonvalue added activities existing in their process (InvSuppl), engagement of the supplier in the green product implementation and green development process of the company (EngParticip). Table 6 reveals the mean and the standard deviation of these five items and Figure 5 shows that more than 60% of the representatives of the participant companies agreed or strongly agreed that green supplier development is an essential determinant for green innovation.

The descriptive statistics of the respondents' perception on the importance of technological factors' impact on green innovation revolves around the following item: the complexity of green practices more specifically: the difficulty of learning green practice (DiffLearnGP), the difficulty in sharing the knowledge of green practice (DiffShareGP), the fact that the green practice needs much expertise (DiffUseGP). Table 7 discloses the mean and the standard deviation of these three items and from Figure 6 it can be observed

that about 45% of the respondents did not agree with the fact that green practices are complicated to learn, more than half of the participants did not think that green practices are difficult to share, and about 60% of the respondents did not generally feel that green practices need much expertise in order to put them to use. Overall, a majority of the respondents had not found green practices to be complex.

**Table 6.** Descriptive analysis for the variables of green supplier development.

|  | N | Mean | Std. Deviation |
|---|---|---|---|
| CommSupplSust | 182 | 4.15 | 0.868 |
| CoopLimSuppl | 182 | 4.70 | 0.882 |
| ChoiceDevSuppl | 182 | 4.49 | 0.819 |
| InvSuppl | 182 | 4.03 | 0.861 |
| EngParticip | 182 | 4.54 | 0.880 |

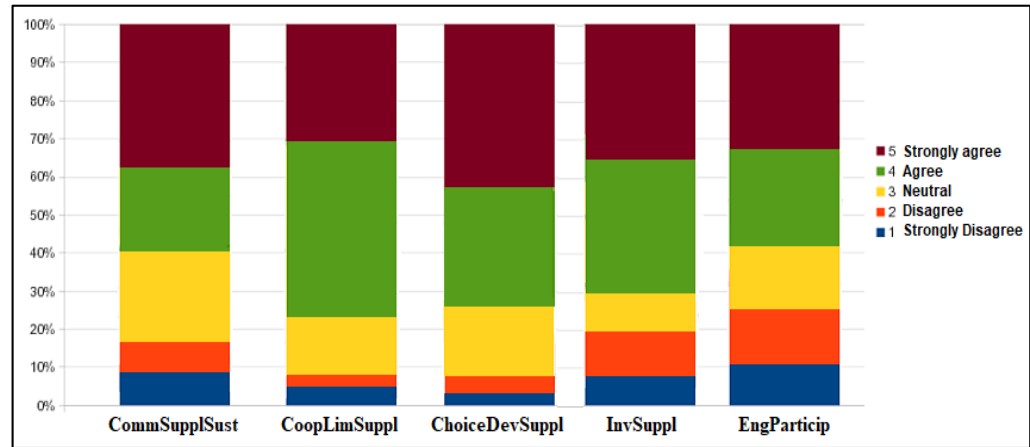

**Figure 5.** Points of view from the respondents regarding the contribution of green supplier development to green innovation.

**Table 7.** Descriptive analysis for the variables of the complexity of green practice.

|  | N | Mean | Std. Deviation |
|---|---|---|---|
| DiffLearnGP | 182 | 2.50 | 1.14 |
| DiffShareGP | 182 | 2.14 | 1.03 |
| DiffUseGP | 182 | 2.00 | 1.01 |

*4.2. Factor Analysis*

This section discusses the factor analysis that was carried out to decrease the number of unreliable factors in the study, making use of the approach called the principal component extraction with the aim of Eigenvalues being higher than 1 (notable as the factors are regarded as trustworthy). The Varimax rotation approach was implemented, and given the sample dimension, the factors that indicated a factor loading higher than 0.50 were kept. Seven distinct factors were recognized for the model all with at least three items. If an item indicated a factor loading lower than 0.50 or it had high cross loading with more than one other factor (variable/construct), the respective item was removed from the model. The items belonging to the compatibility and relative advantage factors were removed because they either recorded insubstantial factor loadings (much lower than 0.50), or had large cross loadings with other constructs (such the ones belonging to green training or green supplier development). This is the reason why they were not present in the previous descriptive analysis. Within the remaining five variables the subsequent items were erased from the econometric framework since they recorded factor loadings below the generally

accepted significance level (0.50): TechAssGov, PredCompet, FreqCustomPref, SuffTrain, CoopLimSuppl, InvSuppl, DiffShareGP.

**Figure 6.** Points of view from the respondents regarding the influence of complexity on the evolution of green innovation.

Table 8 within the next section reveals all five factors which had Eigenvalues higher than 1 (along with the rest of items): government involvement in the development of green innovation (with its two remaining items), the influence of the customers and of the competitors on the company's green innovation (with its three remaining items), the green training input on green innovation (along with the four items), the green supplier development within the company (with the three items) and the complexity of green practices (with its two remaining items). Altogether these five factors total 85% of the entire variance of all the factors in the research, a level regarded as convenient for variance. The variables "green training" and "the influence of the customers and of the competitors on the company's green innovation" were estimated at 27.4% and 21.5% when explaining the total variance, consequently they have a considerable role.

*4.3. Logistic Regression*

The hypotheses were checked with the procedure of logistic regression. The dependent variable was represented by the development of green innovation within the firm (evaluated with Yes or No). The independent variables were exemplified by the composite values estimated with the help of factor analysis. For these variables, normality tests were executed and the outcomes revealed that they were normally distributed. After concluding the process of logistic regression, a fundamental feature of the data that required further analyze was multicollinearity. An exemplary situation is where a strong correlation between the dependent variables and the independent variables exists and a very weak correlation between the independent variables.

The methods which are frequently applied when examining multicollinearity of the independent variables are tolerance and variance influence factor (VIF). A large value of tolerance and a small degree of VIF indicate the advantageous low level of multicollinearity. The recommendations accepted within the literature are a tolerance higher than 0.1 and VIF lower than 10 [52].

The way to diminish the negative influence of collinearity is to remove the items that recorded a high multicollinearity with other factors. In this study, there no high level of collinearity and, therefore, none of the factors had to be eliminated from the model. All

outcomes for tolerance and VIF for the model's independent variables are presented within Table 9.

**Table 8.** Factor Analysis Results.

| | Component | | | | |
|---|---|---|---|---|---|
| | **1** | **2** | **3** | **4** | **5** |
| TrainProc | 996 | | | | |
| ExistTrain | 970 | | | | |
| HighLevContTrain | 952 | | | | |
| EfficTrain | 921 | | | | |
| PressCustomEnvPerf | | 882 | | | |
| CustomCareEnvin | | 860 | | | |
| PredCustom | | 835 | | | |
| CommSupplSust | | | 872 | | |
| ChoiceDevSuppl | | | 840 | | |
| EngParticip | | | 826 | | |
| FinSupGov | | | | 796 | |
| EnvRegGov | | | | 763 | |
| DiffLearnGP | | | | | 757 |
| DiffUseGP | | | | | 732 |
| Initial Eigenvalue | 4.829 | 3.651 | 3.117 | 2.481 | 1.874 |
| Variance | 27.458 | 21.557 | 16.273 | 11.782 | 7.523 |

Extraction Method: Principal Component Analysis; Rotation Method: Varimax with Kaiser Normalisation.

**Table 9.** Collinearity Results.

| Independent Variable | Collinearity Results | |
|---|---|---|
| | **Tolerance** | **VIF** |
| Government support | 0.261 | 3.671 |
| Influence of the customers and of the competitors | 0.322 | 4.016 |
| Green Training | 0.595 | 5.484 |
| Green supplier development | 0.420 | 5.470 |
| Complexity of green practices | 0.742 | 1.963 |

The process of logistic regression measures the log likelihood value of all independent variables applying the Enter technique, with the dependent variable taking the Yes or No values. Table 10 presents to what extent the variation in the dependent variable can be described by the model ($R^2$ within the multiple regression). It reviews the properties regarding the Goodness of fit of the model consisting of the findings of the Cox and Snell R Square and Nagelkerke R Square methods, both being tests of delineating the explained variation [53,54]. The model from this study displayed an explained variation in the dependent variable which ranged from 74.9% to 81%, according to the Cox and Snell $R^2$ method and Nagelkerke $R^2$, respectively. Statistically, the relationship can be perceived as solid because R has a value between 0 and 1 and the higher the values the more intense the model fit.

**Table 10.** The Goodness of Fit for the researched model.

| Step | -2 Log likelihood | Cox& Snell $R^2$ | Nagelkerke $R^2$ |
|---|---|---|---|
| 1 | 12.334 | 0.749 | 0.810 |

Taking into account the outcomes of these tests, one can assume that the goodness of fit of the model is reliable. The next phase was to investigate the correlation between the dependent variable and the independent variables and to find out if there was a meaningful link between the variables. Table 11 summarizes the results of the regression and one can see that from the five independent variables, two of them held a deep relationship with the green innovation development process (the independent variable). These two variables were: green training and the interest of the customers on green issues. It is interesting to analyze the orientation and width of the correlation within the original and exponential coefficients. The Wald test (Wald column) was applied to determine the statistical significance of the independent variables. The statistical significance of the test was retrieved from the "Sig." column and the interpretation was that significant variables should indicate values lower than 0.05 within this column. Table 10 reveals that "green training" ($p = 0.015$) and "the interest of the customers on green issues" ($p = 0.000$) counted substantially for the model, however, the other three variables did not. Both variables mentioned were positively correlated with the green innovation process (revealed by the sign of the B coefficient). Particularly, the probability of developing green innovation within the company was higher for firms that developed green training programs and that took into account the interest of customers in green issues. The outcomes of the logistic regression process rely on odds of happening. For instance, for an additional unit of growth in green training the odds of developing green innovation expand by a factor of 1.814 (B column). The concept of odds ratio exemplifies the ratio of the probability of that event happening over the probability of that event not happening (P (success)/P(failure)). Looking at the findings, holding the other variables with a fixed value, for each one unit of growth in the green training the odds of developing green innovation within the organization expand by 462%. In the logistic regression, operating the figure of Exp(B) minus 1 delivers the percentage change in odds [55].

**Table 11.** Results of the logistic regression.

|  | B | S.E. | Wald | df | Sig. | Exp(B) |
|---|---|---|---|---|---|---|
| Government support | 0.152 | 0.224 | 0.518 | 1 | 0.639 | 1.202 |
| Influence of the customers and of the competitors | 1.481 | 0.302 | 8.741 | 1 | 0.000 * | 4.804 |
| Green Training | 1.814 | 0.507 | 0.832 | 1 | 0.015 * | 5.626 |
| Green supplier development | 0.087 | 0.457 | 0.132 | 1 | 0.835 | 1.019 |
| Complexity of green practices | 1.748 | 0.731 | 4.279 | 1 | 0.634 | 1.137 |
| Constant | 0.378 | 1.353 | 0.156 | 1 | 0.942 | - |

* $p < 0.05$.

The outcomes of the regression investigation confirmed that the two elements which hold a statistically relevant correlation with green innovation development are: "green training" and "the interest of the customers in green issues"; H2 and H3 are confirmed. The other five factors did not influence in a statistically significant way the evolution of green innovation and consequently the corresponding five hypotheses were not confirmed as shown in Table 12. Even though five of the formulated hypotheses were not confirmed, the two which were confirmed underlined the importance of two current issues for 2021. The first one was the significance of green training, of having managers and employees educated in green practices and sustainability, personnel able to promote and apply corporate engagement strategies which could transform the mobile commerce firms into more sustainable companies committed to ecological progress. The second confirmed hypothesis was also related to a current notable issue in the sense that consumers are becoming more and more educated about green business practices and they are willing to spend more when buying from a green innovative company which in return can encourage mobile commerce firms to invest in this direction.

**Table 12.** Statistical support of hypotheses following the logistic regression.

| Study Hypotheses | Confirmed? |
|---|---|
| H1: More solid support from the government positively affects the chances for green innovation development within mobile commerce companies. | **No** |
| H2: Mobile commerce firms that are involved within a business environment strongly influenced by competitors and customers are more likely to develop green innovation. | **Yes** |
| H3: Mobile commerce companies that develop solid green training programs are more likely to enhance green innovation. | **Yes** |
| H4: Mobile commerce firms that strengthen green supplier development programs are more likely to advance green innovation. | **No** |
| H5: Green innovation's perceived complexity negatively affects green innovation development for mobile commerce firms. | **No** |
| H6: Green innovation's perceived compatibility positively influences the green innovation implementation process within mobile commerce firms. | **No** |
| H7: Green innovation's perceived relative advantage positively impacts green innovation development for mobile commerce firms. | **No** |

The aftermath of the study strengthens the idea that companies that reinforce green training programs with frequency of such trainings, held by specialists, and that can be effectively applied, are more likely to enhance the green innovation of the firm. Simultaneously, expanding a solid relationship with clients on green issues and addressing the environmental concerns of the customers increases the opportunities for green innovation within the respective company.

## 5. Conclusions

The paper provides valuable contributions for academia and business. It is worth mentioning that within the European Innovation Scoreboard for 2020, Romania had the weakest score from the EU with a 68% gap from the EU average score. The main problems for Romania, revealed by the EU study, were human resources and investments in R&D [56]. These results match part of the outcomes of this study, underlining the importance of competent and trained personnel, and emphasize the fact that policy makers should establish more programs and more opportunities for lifelong learning and for HR professional development including topics such as sustainable development and environmental management.

Most of all, from a theoretical point of view, the econometrical model that has been advanced in this research is authentic and has not been used in other studies in Romania. This original article for the Romanian academic field investigated how mobile commerce firms boost green innovation through the development of green training and green supplier development programs. The study adds to existing literature on issues of environmental operations management and green HR management because it analyzes how different factors impact green innovation within a mobile commerce firm's context. The accomplishment from these analyses is that from the seven constructs examined, two are highly correlated, indicating that when a company is committed to green training programs and to the interest shown by customers in green issues, the respective mobile commerce firm can achieve a higher level of green innovation. The results are useful from a practical point of view, drawing the attention of firm managers to the fact that such employee education and training programs are particularly important, because the employees' familiarity with green practices will enable more effective implementation of sustainable development in their respective firms and therefore boost green innovation. At the same time, companies interested in increasing their green innovation need to communicate well and frequently with their clients on green problems so that the expectations of both sides can be met.

This article includes some limitations as well. The one which is taken into consideration for improvement within future research regards the extent of the sample (182 firms) which is not low, but the procedure of logistic regression confers appreciable outcomes for more independent variables when the sample is larger than two hundred. Nevertheless, in this example, meaningful results were reached for two out of seven variables, which is an adequate result for this type of investigation. The approach of gathering information from more than 180 mobile commerce companies in Romania presented some challenges related to the preoccupation with sustainability and the knowledge of green issues of the respondents within Romanian mobile commerce firms. Another limitation could consist in the fact that the sample assembled Romanian companies, and consequently it is less expected that the results could be applied to foreign companies [57], but there could be an opportunity to develop a further study involving companies from other countries from Southeast Europe, states with similar economies and businesses approaches.

For future research the authors plan to create a larger and more diverse sample that would augment the study of the econometric model, for instance using other purchasing methods, (e.g., shopping through a desktop computer not mobile phone) to see how they compare to the mobile commerce area.

**Author Contributions:** Conceptualization, V.M.D. and C.V.; methodology, V.M.D. and A.B.; software, V.M.D. and F.Z.; validation, V.M.D., C.V. and F.Z.; formal analysis, V.M.D. and A.B.; resources, A.B., C.V. and F.Z.; writing—original draft preparation, V.M.D., C.V., A.B., and F.Z.; writing—review and editing, V.M.D., C.V., A.B., and F.Z.; visualization, V.M.D.; supervision, V.M.D. All authors have read and agreed to the published version of the manuscript.

**Funding:** This research received no external funding.

**Data Availability Statement:** Not applicable.

**Conflicts of Interest:** The authors declare no conflict of interest.

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
