# Peer review of "The Influence of Romanian Mobile Commerce Companies on Developing Green Innovation"

_sustainability, doi:10.3390/su131810075_

Round 1
Reviewer 1 Report
What do you mean by: mobile commerce companies?
On page 2, line 62 you talk about - online commerce businesses and e-commerce sector
Line 80 (with the mobile commerce as a sub-sector) - please write at the very beginning what you are actually researching. It’s only after page 2 that I understood that you are researching online retail with possibility to buy via smart phones.
I don’t understand. If you are researching mobile retail – of what kind of innovation are we talking about since they are not the producers, rather the polluters filling the last line in the journey of a product to customer.
The 2 paragraphs from literature review would better fit in introduction where you show clear numbers that retail through mobile devices is becoming greater then desktop purchases.
Page 3
Therefore (here comma is missing) according to Esmaeilpour and Bahmiary (2017) it is more and more recom- 125
mended for mobile commerce companies to focus on products that are compatible with 126
the environment because the consumers’ welcome for purchasing such products is more 127
and more current [19]
This constatation in my opinion is valid for all e-commerce not just mobile
Where is this Hypothesis 1 coming from?
The government support for mobile commerce firms is 132
positively related to the implementation of green innovation (Hypothesis 1)
You didn’t talk about government role before. And again, I don’t understand what innovation you are talking about? Are those more eco friendly products in the shop? But in that case, the innovation is done by the manufacturer and not the retailer.
As a result, it is firmly assumed that 147
green training in eco-design, LCA, recycling/reusing, and waste elimination will point 148
the way to green innovation advancement [27,28].
GT is absolutely welcome even among retailers, but eco-design, waste elimination ,…. Is done by the manufacturer.
Page 4.
Rephrase all your hypotheses 5-7 to the context you are exploring. I still don’t see the m-commerce innovation, rather you describe in previous paragraphs that the manufacturers make more green products in cooperation with their supply chain partners. The only thing m-commerce, as you call it, can decide which eco friendly products it will display on their platform.
Page 5
Line 211-237
It is not usual to describe prescript values. Rather you show your results and then comment is it in acceptable range. Now we know nothing about your results.
Page 6, figure 2
This must be some peculiarity of Romania, because in most countries’ governments do not subsidize in any way retailers. I think you have to explain it better.
Details on measurement methods are missing.
Page 10, Table 7
Instead of abbreviations put the whole name of the researched independent variable as you did in Table 8.
Again, the comment, that we don’t know how you measured it.
Usually, the measurement instrument is described in the methodology part or the whole measurement instrument with sources is put in the Appendix. Please fix that so we know what you measured.
Author Response
First of all we would like to express our gratitude for all the feedback provided in order to further expand our paper. Your input and recommendations were highly acknowledged and taken into account by the authors. We implemented your comments as follows:
- The theoretical background of the paper has been developed by adding two paragraphs with the problem statement – paragraphs marked with violet in the introduction part (page 2). We tried to give to the paper a clearer background that our research is based on mobile commerce companies and their respective innovation (trying to eliminate the overlap with online commerce in general);
- Two longer paragraphs were added (page 3 in the middle and in the bottom – continuing on the top of page 4 - marked with red) in order to give to the hypotheses (and consequently to the variables – government support etc) a deeper discussion and more support. These paragraphs also clarify the innovation we were investigating. In this sense we also inserted an extra table with variables/ description of variables and hypotheses (table 1 from page 5) and a research diagram (figure 1 from page 6);
- In order to clarify the measurement instrument à within the methodology - in the middle of page 6 - we added a paragraph explaining that the data collection process was founded on a questionnaire and its quantification tool was the 5 point Likert-scale analyzing the influence of all factors on implementing green innovation within the company (with 1 considering that the factor has the smallest influence and 5 - the biggest influence);
- At the end of the discussions’ section (page 14 – lines 3 to 12, before table 12) we added a longer paragraph (with violet) stating the importance of the validation of the two hypotheses bringing into discussion a few important issues for today’s economy in order to make this section more solid.
Thank you again for all your guidance and advice.

Reviewer 2 Report
The paper entitled” The Influence of Romanian Mobile Commerce Companies on Developing Green Innovation” deals with actual and very interesting topic.
However, I have the following comments that hopefully help the authors improve their paper:
- Overall, the paper lack’s structure and a clear flow. It requires a complete restructuring of the sections, as well as contents both in topic and methodology.
- The structure (outline) of the paper could be given at the end of the introductory chapter 1
- I suggest that the authors add a research method diagram. This will provide a snapshot of the research steps followed and will help the reader in a clearer understanding of the paper.
- The literature review is mixed with hypotheses formulation. I suggest to the authors a section dedicated to literature review where should analyse the existing works in the way to show the gap in the literature compared to this work. For instance, it would be better if authors can have a table comparing the closely related works on various dimensions and clearly showing the contribution of the paper.
- In relation to hypotheses formulation, the assumptions of the approach proposed deserves a deeper discussion. The hypotheses are not critically discussed. Furthermore, I recommend that the authors include a diagram of your approach in which they could established the relationships among several hypotheses in a summary way.
- In section 4 descriptive analysis, hypothesis testing and results the discussion looks weak and these need to be elaborated.
- Figures are very small and hard to read.
- The authors should convince the readers of this journal, that their contribution is so important. In this context, these issues deserve a deeper discussion: What are the implications for theory and practice? What are the managerial implications from this work? How decision or policy makers could benefit from this study.
- How the results of this study can be generalized to other countries. The main contribution of this work should be compared with other similar empirical studies.
- As usual a final thorough proof-reading is recommended.
I encourage the authors to think along those questions and to develop this work further along those lines.
Author Response
First of all we would like to express our gratitude for all the feedback provided in order to further expand our paper. Your comments and recommendations were highly acknowledged and taken into account by the authors. We implemented the changes you suggested as follows:
- The theoretical background of the paper has been developed by adding two paragraphs with the problem statement – paragraphs marked with violet in the introduction part (page 2). We tried to give to the paper a clearer background by doing this;
- The structure (outline) of the paper was compiled at the end of the introduction (in the bottom of page 2 - marked with red);
- Two longer paragraphs were added (page 3 - marked with red) in order to give to the hypotheses a deeper discussion and more support. In this sense we also inserted an extra table with variables/ description of variables and hypotheses (table 1 from page 5) and a research diagram (figure 1 from page 6);
- At the end of the discussions’ section (page 14 – lines 3 to 12, before table 12) we added a longer paragraph (with violet) stating the importance of the validation of the two hypotheses bringing into discussion a few important issues for today’s economy in order to make this section more solid;
- We enlarged the figures from section 4 (discussions) so that they would be more visible;
- The contributions of the paper were expanded by adding a paragraph (with violet - page 14 – in the bottom of the page) about the interest that policy makers for innovation could have in this topic/ paper, and overall the conclusions’ section treated separately the importance of this paper for theory, practice (businesses), managers, policy makers – changes marked with red in the conclusions’ section;
- At the end of the conclusions we added a paragraph (page 15 with red) about how the results of this study could be generalized to other countries.
Thank you again for all your guidance and support.

Reviewer 3 Report
The Influence of Romanian Mobile Commerce Companies on Developing Green Innovation
The aim of the paper is to identify the factors that determine the development of green innovation within 182 Romanian mobile commerce companies from Bucharest.
Highlights of the paper per category:
Originality/Novelty: The question put into discussion by the paper is interesting combing elements of innovation with those related to environment and sustainability. In the same time not many similar studies were created so far for the Romanian businesses so this aspect gives an advanced level of originality to the article.
Significance: The results are accurately interpreted and they bring significance to the researched field. In the same time the conclusions underline the importance of the study but also its limitations. The hypotheses are very clearly answered because of good statistical calculations and interpretations.
Quality of Presentation: The article is written in a logical manner, following a structure with hypotheses that are being backed up by a good literature review part. The manuscript should be even more sufficiently supported by evidence or proper references to work done elsewhere. For this purpose authors could downloads the following published article: https://doi.org/10.3390/su11010089, http://anale.steconomiceuoradea.ro/volume/2018/AUOES-1-2018.pdf
The data and analyses presented prove that the authors have good skills in using statistical software and always following the hypotheses and coming back to them in order to support the objectives of the paper.
Scientific Soundness: The study provides analyses performed with good statistical software and the methods and tools are rigorously presented and applied so that the model can be thoroughly explained and supported.
Interest to the Readers: The subject of the paper is appealing and the quality of the presentation invites readers in many areas to become interested in the paper. For the conclusions’ part the authors should provide more information regarding future studies.
Overall Merit: The paper has a high degree of originality and it can stand as a potential inspiration for other researchers to study how sustainable development and innovation can complement each other.
English Level: The English language is correctly used and coherently expressed.
Author Response
Thank you very much for all your feedback. Your appreciation and recommendations were taken in high regard by all authors. We implemented the changes you suggested by adding the very interesting sources advised in your review report – you can find them in the bibliography at no. 22 and 29 and in text marked with red at page 4 (first seven lines and in the middle of the page). Thank you again for all your support.

Reviewer 4 Report
Regarding the paper with the title "The Influence of Romanian Mobile Commerce Companies on Developing Green Innovation" I have few remarks:
- Improve paper “theoretical background" with the discussion of problem statement in the introduction part.
- Improve the "conclusion” in terms of writing the significance of this study and explain in the “discussion part” why the validation of only two hypotheses is enough for this study from the seven formulated hypotheses.
- Clarify the paper contributions more specifically.
Author Response
First of all we would like to thank you very much for all the feedback given for further developing our article. Your comments and recommendations were highly acknowledged and taken into consideration by the authors. We implemented the changes you suggested as follows:
- The theoretical background of the paper has been developed by adding two paragraphs with the problem statement – paragraphs marked with violet in the introduction part (page 2)
- In the discussions’ section (page 14 – lines 3 to 12, before table 12) we added a paragraph (also with violet) stating the importance of the validation of the two hypotheses bringing into discussion a few important issues for today’s economy in order to make this section more solid.
- The contributions of the paper were expanded in the conclusions by adding a paragraph (with violet - page 14 – in the bottom of the page) about the interest that policy makers for innovation could have in this topic/ paper, and overall the conclusions’ section treated separately the importance of this paper for theory, practice (businesses), managers, policy makers – changes marked with red in the conclusions’ section.
Thank you again for all your guidance and support.

Round 2
Reviewer 1 Report
Authors clarified the flaws in the article. It’s still not a great paper but is acceptable.
Reviewer 2 Report
The manuscript has significantly improved as compared to the previous version. Indeed, the authors tried to improve it, and the main weaknesses are solved. Thus, in my opinion, the manuscript is recommendable for publication.